# Evaluation of Anti-Angiogenic Therapy Combined with Immunotherapy and Chemotherapy as a Strategy to Treat Locally Advanced and Metastatic Non-Small-Cell Lung Cancer

**DOI:** 10.3390/cancers16244207

**Published:** 2024-12-17

**Authors:** Mahmoud Abdallah, Rick Voland, Malcolm Decamp, John Flickinger, Toni Pacioles, Muhammad Jamil, Damian Silbermins, Mina Shenouda, Matias Valsecchi, Arvinder Bir, Yousef Shweihat, Juan Bastidas, Nepal Chowdhury, Yury Kachynski, Howide Eldib, Thomas Wright, Ahmad Mahdi, Jowan Al-Nusair, Kemnasom Nwanwene, John Varlotto

**Affiliations:** 1Department of Oncology, Edwards Comprehensive Cancer Institute, Marshall University, Huntington, WV 25701, USA; abdallahm@marshall.edu (M.A.); pacioles@marshall.edu (T.P.); jamil@marshall.edu (M.J.); damiansilberminsm.d@uhswv.com (D.S.); shenoudam@marshall.edu (M.S.); matiasvalsecchim.d.m.s@uhswv.com (M.V.); arvinderbir@uhswv.com (A.B.); shweihat@marshall.edu (Y.S.); juan.bastidas@st-marys.org (J.B.); nepal.chowdhury@st-marys.org (N.C.); kachynski@marshall.edu (Y.K.); eldib@marshall.edu (H.E.); nwanwene@marshall.edu (K.N.); 2Department of Ophthalmology, University of Wisconsin, Madison, WI 53705, USA; rvoland25@gmail.com; 3Division of Cardiothoracic Surgery, University of Wisconsin School of Medicine and Public Health, Madison, WI 53726, USA; decamp@surgery.wisc.edu; 4Department of Radiation Oncology, University of Pittsburgh Medical Center, Pittsburgh, PA 15219, USA; flickingerjc@upmc.edu; 5Department of Internal Medicine, Marshall Health, Huntington, WV 25701, USA; wright210@marshall.edu (T.W.); mahdi@marshall.edu (A.M.); alnusair@marshall.edu (J.A.-N.)

**Keywords:** lung cancer treatment, anti-angiogenic therapy, immunotherapy, chemotherapy, non-small-cell lung cancer (NSCLC)

## Abstract

Around 25–30% of non-small-cell lung cancers (NSCLC) present with locally advanced, unresectable disease where treatment with concurrent chemo/radiation followed by durvalumab remains the standard of care. However, only about one third of patients are alive without progression at 5 years. Therefore, there is a great need for improvement. Due to flaws of the past trial designs, the use of anti-angiogenic agents with radiation have been abandoned for Stage III NSCLC. We review how to use anti-angiogenic therapy safely in the locally advanced setting and discuss its expected beneficial effects. We also will review the how combined chemotherapy, anti-angiogenic therapy and immunotherapy (AIC) can be used in the metastatic setting and how it can be used as a strategy of choice for patients with liver/brain metastases as well as those with mutations that are considered to be less responsive to immunotherapy, such as, STK11, KRAS, and KEAP1. Additionally, we review trials and discuss the benefits of using AIC therapy in patients with metastatic EGFR mutations in the thirdline setting. Innovative trial designs are proposed using AIC therapy in the locally advanced setting as well as for the upfront treatment of patients suffering from brain metastases.

## 1. Introduction

This article will review the tumor micro-environment and help the reader to understand how anti-angiogenic therapy (AA) can overcome the tumor’s local immunosuppressive effects and thereby improve the efficacy of immunotherapy (IO). Additionally, the success and failures of past uses of AA strategies with or without chemotherapy (CT), radiation therapy (RT), and/or IO is reviewed so that we can hypothesize how to improve the outcomes for locally advanced or metastatic NSCLC.

The tumor micro-environment (TME) consists of blood and lymphatic vessels, various stromal cells, and resident and infiltrating immune cells, all of which are ensconced in an extracellular matrix. In comparison with nonmalignant tissues, each component of the TME is abnormal and heterogeneous, and this abnormality fuels tumor progression and treatment resistance [1]. Normalizing components of the TME could improve the outcomes of patients treated with standard and emerging anticancer therapies, including CT, RT, molecularly targeted agents, and IO [2].

We feel that the combination of AA therapy, IO, and cytotoxic therapy may be beneficial for non-small-cell lung cancer (NSCLC) in the setting of metastatic disease without targetable mutations, brain metastases, locally advanced disease, and for recurrent EGFR-mutated (EGFRmut) disease. We herein review the clinical research for locally advanced and metastatic NSCLC and discuss how we can improve our therapeutic armamentarium in these four areas.

## 2. Materials and Methods

For this review article, the most recent treatments and outcomes for NSCLC were compiled via an updated literature search investigation using https://meetinglibrary.asco.org/, https://www.jto.org/action/doSearch?type=quicksearch&text1=world+lung+meeting+abstracts&field1=AllField, https://www.ncbi.nlm.nih.gov/pubmed, and https://www.esmo.org/meetings/past-meetings on-line databases for recent articles and/or abstracts (accessed on 21 September 2024).

The PubMed/MEDLINE database (1990–2024) was queried using the investigational strategy immunotherapy (nivolumab, pembrolizumab, durvalumab, ipilimumab, cemiplimab, and atezolizumab; “locally advanced” and “non-small cell lung cancer”) and (“outcome”) brain metastases and anti-angiogenic therapy (“bevacizumab, ramucirumab, ivonescimab”) for updated information regarding treatment and outcomes of non-small-cell lung cancer with metastatic or locally advanced presentations.

### 2.1. Background—The Tumor Microenvironment

The concept that growing tumors have a rich vascular network first arose well over 100 years ago through observations by notable scientists such as Virchow and was strengthened by the seminal work of Ide [3] and later Algire, who confirmed the importance of an abundant blood supply to tumor growth [4], forming the foundation for Dr. Judah Folkman’s seminal paper in 1971 in which he suggested that the identification of key molecular players driving tumor angiogenesis could result in effective strategies to inhibit it and hence “starve” a tumor to death [5]. It has now been established that hypoxia is a hallmark of solid tumors and that this in turn drives the production of angiogenic factors, including VEGF [6]. The VEGF family of proteins includes VEGF-A, VEGF-B, VEGF-C, VEGF-D, and placental growth factor (PlGF) as well as the virus-encoded VEGF-E and the snake venom-derived VEGF-F [7]. VEGF-A is the most potent stimulator of angiogenesis and binds to VEGFR1 and VEGFR 2, but most of its deleterious effects for cancer growth occur via VEGFR2 [8]. Therefore, the most potent angiogenic inhibitory agents in the treatment of NSCLC either block VEGF-A directly (bevacizumab [9]; ivonescimab [10]) or directly block VEGFR2 (ramucirumab [11]). Figure 1 includes the VEGF family of proteins and their associated receptors as well and the actions of the angiogenic inhibitors.

At the cellular level, the endothelial cells (ECs) lining tumor vessels have an irregular, disorganized morphology. Mature, stable ECs are connected by adherens junctions, including vascular endothelial (VE)–cadherin. VE–cadherin is a transmembrane receptor, the extracellular domain of which binds to other VE–cadherin molecules on neighboring ECs. The intracellular domain of VE–cadherin attaches to the EC cytoskeleton via the catenin family of proteins, acting as a structural link but also as an effector for downstream molecular signaling. Downstream signaling from VEGF–VEGFR2 interactions promotes contraction of the EC cytoskeleton and weakening of VE–cadherin junction, and hence a loosening of EC associations and EC migration [12]. As a consequence, ECs within tumors are often poorly connected or overlapping, with less VE–cadherin and a branched phenotype with long cytoplasmic projections [13]; these structural aberrations may occur heterogeneously throughout any given tumor during growth, progression, and response to therapy, adding an extra layer of complexity to our understanding of the tumor microenvironment [14].

Among various angiogenic molecules, VEGF is known to have diverse activities in the TME. In particular, the roles of VEGF in systemic and local immunosuppression have been extensively investigated in tumor models (see Figure 2) [15]. Excessive levels of VEGF in the TME induce tumor-associated immunosuppression via at least five distinct mechanisms. First, increased VEGF directly inhibits cytotoxic T-lymphocyte trafficking, proliferation, and effector function. Second, VEGF inhibits DC maturation and antigen presentation, thus hampering T-cell activation and, consequently, diminishing the T-cell-mediated anticancer immune response [16]. Third, high levels of VEGF promote the recruitment and proliferation of immunosuppressive cells, including T_reg_ cells, myeloid-derived suppressor cells (MDSCs), and pro-tumor, M2-like TAMs (tumor-associated macrophages) [17]. Fourth, VEGF promotes angiogenesis that results in an aberrant tumor vasculature, leading to hypoxia and a low pH in the TME, which in turn fosters immunosuppression both locally and systemically. Fifth, as noted in the preceding paragraph, there is endothelial cell disruption, resulting in vascular collapse and diminished blood supply [18].

VEGF and other angiogenic factors, such as angiopoietin 2 (ANG2), modulate the functions of innate and adaptive immune cells towards immunosuppression. VEGF can be produced by tumor cells and immune cells as well as by endothelial cells and stromal cells. These VEGF-stimulated regulatory T (T_reg_) cells, TAMs, and/or monocytes also produce VEGF and ANG2 and thereby engage in paracrine and autocrine VEGF (and/or ANG2) signaling. These downstream effects have been known for decades [19,20]. Among immune cells, T_reg_ cells have been identified as the major source of VEGF in the tumor microenvironment using an in vivo model [21]. Dual anti-VEGF–ANG2 therapy with A2V has been shown to upregulate the expression of adhesion molecules during the window of vascular normalization and can thereby facilitate the accumulation of anticancer T cells within multiple types of tumors in mice. The targeting of VEGF–ANG2 can be enhanced via immune checkpoint inhibition and may represent a viable pathway to target in the future [22].

Synergy between cytotoxic therapy and anti-angiogenic therapy (AA) has been hypothesized. Cytotoxic therapy would kill cancer cells directly, and AA would cause vascular regression and hence deprivation of nutrients to cancer cells [23]. Beyond this, the data have also suggested that CT and RT may also have antiangiogenic effects, directly damaging tumor ECs or preventing recruitment of endothelial progenitor cells and hence enhancing the indirect killing of cancer cells [23]. Moreover, cancer cells may express receptors for VEGF, and thus, AA drugs could directly interfere with their survival or increase their sensitivity to other therapies [2]. Despite the multitude of possible mechanisms for synergy, there is a fundamental paradox that chemotherapy and radiotherapy rely on adequate tumor perfusion. The normalization phenotype helps to explain this paradox, through the maturation and improved functionality of micro vessels [24].

The TME, characterized by hypoxia, a low pH, and a high interstitial fluid pressure, is immunosuppressive and can reduce the effectiveness of virtually all types of anticancer therapies and IO [1]. Therefore, normalizing components of the TME could improve the outcomes of patients treated with standard and emerging anticancer therapeutics [25]. Preclinical studies demonstrated increased tumor infiltration by T cells in response to low doses of an anti-VEGFR2 antibody during the window of normalization, thereby improving dendritic cell maturation and antigen presentation [26,27]. Upregulation of PD-L1 has been demonstrated on both ECs and tumor cells following treatment with anti-VEGFR2 therapy [28]. In fact, anti-angiogenic factors have been shown to increase lymphocyte trafficking across endothelia into tumor deposits [29]. As such, synergy is proposed when combining AA with IO due to the upregulation of PD-L1, T-cell trafficking, and antigen presentation, which is proposed to re-invigorate exhausted/dysfunctional T cells [30].

Compelling preclinical and clinical evidence indicates that anti-VEGF therapy creates a transient window of vessel normalization during which the delivery of oxygen—a radiosensitizer and immunostimulatory—and various therapeutic agents to tumors is improved [2]. IOs can dramatically prolong the survival of patients who have durable responses, but they have generally failed to improve the outcomes of the remaining majority of patients, at the cost of substantial toxicities in many recipients. AA drugs offer only a modest survival benefit of a few weeks to months, with rare durable responses. Herein, we discuss the potential to rationally combine these two approaches to increase patient survival beyond that currently conferred by each approach individually.

### 2.2. Studies in Advanced NSCLC Using Combinations of Anti-Angiogenic Therapy and ICI +/− Chemotherapy (AIC)

#### 2.2.1. Studies of Anti-Angiogenic Agents with Chemotherapy

The use of bevacizumab (a humanized monoclonal antibody directed against VEGF-A) is limited to only patients with non-squamous NSCLC. Bevacizumab was noted to be associated with life-threatening pulmonary hemorrhage in 6 out of 66 patients in a randomized phase II trial of carboplatin and paclitaxel with or without bevacizumab in patients with metastatic NSCLC [31]. Since these toxicities were more common in squamous cell carcinomas, patients with squamous cell carcinoma, gross hemoptysis (0.5 tsp per event), and CNS metastases were not permitted to enter the phase III trial (ECOG 4599), which demonstrated an overall survival (OS) benefit of bevacizumab and chemotherapy (carboplatin/paclitaxel) as compared to the same chemotherapy alone [32]. Interestingly, when bevacizumab was combined with cisplatin and gemcitabine, there was no survival benefit noted as compared to the same chemotherapy in patients with metastatic non-squamous NSCLC [33]. However, the REVEL trial noted that ramucirumab (a fully human immunoglobulin G1 that selectively binds with high affinity to the extracellular domain of VEGFR-2) plus docetaxel not only improved OS in comparison to the same chemotherapy and placebo [34], but ramucirumab’s effect on OS was noted to have the same hazard ratio (HR) for survival independent of histology (adenocarcinoma HR 0.83; squamous cell carcinoma HR 0.88; other non-squamous cell carcinomas HR0.86) [35]. Any grade bleeding events were overall higher in the ramucirumab arm (31.2% NS-NSCLC and 30.8% adenocarcinoma) vs. the placebo group (13.6% NS-NSCLC and 14.2% adenocarcinoma). However, >grade 3 bleeding/hemorrhage rates were similar across the squamous, other non-squamous, and adenocarcinoma sub-groups. In particular, the rates of ≥grade 3 bleeding events were low and similar for patients with squamous cell carcinoma in the ramucirumab (2.5%) and placebo arms (2.9%). Of the 1253 patients in this trial, 12 fatal pulmonary hemorrhages were noted, 6 in each of the randomized arms. Although the REVEL trial proved the safety and efficacy of ramucirumab irrespective of histology, it should be noted that the trial required that all squamous cell carcinomas or centrally located mediastinal masses (<3 cm from the carina) undergo an MRI or contrast-enhanced CT scan within 21 days of randomization to exclude major blood vessel or airway invasion.

#### 2.2.2. Investigations of Anti-Angiogenic Agents and Immunotherapy with or Without Chemotherapy

The phase 3 IMpower150 study evaluated the efficacy of atezolizumab–bevacizumab–carboplatin–paclitaxel (ABCP) or atezolizumab–carboplatin–paclitaxel (ACP) versus bevacizumab–carboplatin–paclitaxel (BCP) in an all-comer patient population with metastatic non-squamous NSCLC (NS-NSCLC). ABCP versus BCP had statistically significant and clinically meaningful progression-free survival (PFS; *p* < 0.001) and OS (*p* = 0.02) benefit regardless of PD-L1 expression and numerical improvements irrespective of EGFR or ALK genetic alteration status [36]. Based on these results, ABCP has been approved in the United States for the first-line treatment of metastatic NS-NSCLC without EGFR or ALK genetic alterations [37]. The European Commission also approved this regimen for first-line treatment of NS-NSCLC and for tumors having EGFRmut and ALK translocations after failure on appropriate tyrosine kinase inhibitor (TKI) therapy [38]. Additionally, in the subset analysis, ABCP was noted to have a PFS benefit as compared to BCP in patients with KRAS mutant and KRAS wild-type tumors [39]. In patients with metastatic non-squamous NSCLC without EGFR or ALK genetic alterations, the ABCP arm was associated with a significantly longer median PFS (8.3 vs. 6.8 months), median OS (19.2 vs. 14.7 months), and response rates (63.5% vs. 48.0%). The incidences of rash, stomatitis, febrile neutropenia, and hemoptysis were higher by less than 10 percentage points among the patients in the ABCP group than among those in the BCP group [36]. With ACP versus BCP, a survival benefit was not observed [36].

In the final IMpower150 analysis with 39.8 months of follow-up, the ACP arm revealed numerical but not statistically significant OS improvement versus the BCP arm. However, there was a continued significant OS improvement with ABCP versus BCP [40].

Recently, it was noted that liver metastases have poor response to PD-1 inhibition and are also the one of the most common sites of progression [41]. Similarly, this relatively poor response was noted in melanoma and NSCLC where liver biopsies noted a reduced CD8+ T-cell density at the invasive tumor margin [42]. The liver may promote immunotolerance through the secretion of IL-10 by liver dendritic cells (DC) upon stimulation. Through IL-10, the liver DCs generated more suppressive CD4(+)CD25(+)FoxP3(+) T regulatory cells and IL-4-producing Th2 cells [43,44]. In subset analyses from prospective, randomized trials, the presence of liver metastases was found to have trend towards a slightly better survival with CT than with either combined IO (nivolumab and ipilimumab, [45]) and combined IO and CT (atezolizumab, carboplatin, and paclitaxel [46]), while those without liver metastases clearly showed a significant survival benefit in both IO groups. Interestingly, when combined with bevacizumab, combined CT and IO (atezolizumab) was associated with significantly better OS with or without liver metastases than the same CT (carboplatin and paclitaxel) combined with bevacizumab in ImPower 150 [47]. Additionally, this combined ABCP regimen was associated with a delayed time to development of brain metastases as compared to BCP. Interestingly, post hoc analysis of this trial noted that the arms containing bevacizumab had a lower rate of brain metastases than the atezolizumab/CT arm (7.0% with atezolizumab/bevacizumab/chemotherapy and 6.0% with bevacizumab/chemotherapy as compared to 11.9% in the atezolizumab/chemotherapy arm) [47].

Using a randomized phase 2 design, the Lung-MAP nonmatch substudy (S1800A) assessed ramucirumab and pembrolizumab (RP) or investigator’s standard of care (67% received ramucirumab and docetaxel; the others received single-agent chemotherapy) in patients who had failed previous treatment with IO and platinum-based CT. OS was significantly improved with RP (14.5 months) as compared to the standard of care (11.6 months), and RP had much less grade 3+ toxicity (42% vs. 60%). Importantly, this therapy was noted to yield OS benefits in an unplanned subset analysis in patients with difficult-to-treat mutations that are usually recalcitrant to immunotherapy, including p53, CDKN2A, KRAS, STK11, and KEAP1 [48]. The success of RP led to the Pragmatica Trial (NCT05633602), which uses the same randomized design in an expanded phase 3 trial.

Ivonescimab is a novel, bi-specific antibody that blocks the binding of PD-1 to its ligand (PD-L1) and also inhibits the binding of VEGF-A to its main receptor (VEGFR2). The HARMONi-2 trial was conducted in China and randomized patients to ivonescimab or pembrolizumab for patients with advanced PD-L1+ NSCLC (without EGFR mutations of ALK rearrangements) in the first-line setting. After 8.7 months of follow-up, ivonecimab improved median PFS from 5.8 months to 11.14 months. Subgroup analyses demonstrated a consistent benefit of ivonescimab for PFS in PD-L1 low (1–49%), PD-L1 high (50%+), as well as squamous and non-squamous histologies [49].

In summary, the combination of IO and AA therapy is an effective therapy that yields a PFS and OS benefit as compared to BCP and may have greater efficacy in those patients with liver metastases and those tumors with mutations that are typically difficult to treat with IO. Additionally, this combined therapy may also have activity in brain metastases.

### 2.3. Can Combing Chemotherapy, Immunotherapy and Anti-Angiogenic Therapy Reduce Intracranial Metastatic Tumor Burden?

To understand how to integrate systemic therapies into the management of NSCLC patients with brain metastases, we must first understand the limitations of brain radiotherapeutic techniques.

#### 2.3.1. The Efficacy and Limitations of Radiation for Brain Metastases

For patients with extensive intracranial brain metastases, the treatment of choice has been whole-brain radiation therapy (WBRT). Even with the use of memantine, cognitive function failure was noted in 53.8% as compared to 64.9% with placebo at 24 weeks [50]. In patients eligible for stereotactic radiosurgery (SRS), a recent randomized study noted that compared to receiving SRS alone, patients receiving WBRT and SRS showed much greater cognitive deterioration at 3 months (91.5% vs. 63.5%) and worse quality of life despite having better intracranial control and a similar overall survival [51]. Although a recent phase III study showed that whole-brain radiation therapy delivered with hippocampal avoidance and memantine showed less neurocognitive failure than conventional whole-brain radiation therapy and memantine, neurocognitive failures in the hippocampal-avoidance arm were still greater than 50% at 6 months [52].

Although SRS or stereotactic radiation therapy (SRT) alone are generally used for more limited presentations of brain metastases (four or fewer tumors with all lesions < 3.5 cm), there have been concerns of increased risks of radionecrosis (>35%) when patients receive such focal radiation techniques with IO [53,54]. Although one recent series did not reveal an increased risk of radiation-related necrosis when patients with melanoma brain metastases received both SRS and IO, it did note that even in modern SRS series, larger median target volumes were associated with enhanced risk of radionecrosis [55].

Four prospective, randomized trials have compared SRS to WBRT and SRS [51,56,57,58]. Although WBRT in addition to SRS resulted in higher rates of intracranial control both locally and distally, there was no survival benefit in any of the trials. The results from the SRS arms of these four prospective, randomized trials are shown below in Table 1. Rates of local and distal intracranial control in the SRS only arms ranged from 67.0–72.8% and 36.3–69.9%, respectively. Rates of symptomatic radionecrosis ranged from 1–6.67% in these four studies that were conducted prior to the approvals of IO in the first- or second-line settings. Most patients in these four prospective, randomized trials had NSCLC. However, despite the heterogeneous primaries in these four studies, the local control rates of approximately 70% in these trials pertain to our suggested trial design because it has long been established that the high local control rates associated with SRS are independent of tumor histology [59]. Furthermore, as noted below, despite varying percentages of NSCLC in these trials (52–72.1%), similar local control rates were noted with SRS. Additionally, local control rates (67–72.8%) and distal control rates (36.3–69.9%) need improvement.

Therefore, there is great need for reducing the metastatic intracranial burden in order to allow for conversion of patients requiring whole-brain radiation to appropriately receive SRS/SRT alone and for size reduction of the metastatic lesion to permit higher doses of radiation and a lower risk of complications even when SRS/SRT is given alone. Because the SRS doses are increased inversely with lesion size [60], reducing the size of metastases would not only allow for more effective higher doses, but the size reduction would greatly diminish the volume of normal brain outside of the target receiving lower doses associated with symptomatic brain necrosis in patients undergoing SRS for brain metastases as well as other intracranial targets [61,62,63]. Furthermore, it is hoped that upfront and continued IO can also improve intracranial control.

#### 2.3.2. The Role of Systemic Therapy in the Treatment of Brain Metastases from NSCLC

There is evidence that systemic agents may be able to cross the blood–brain barrier and result in intracranial responses in NSCLC patients. As noted in the Pacific Trial, The PD-L1 inhibitor durvalumab decreased the incidence of brain metastases when it followed chemo/radiation for patients with locally advanced NSCLC patients as compared to placebo in both the initial report on progression-free survival (PFS) and its later report on overall survival (OS) [64,65]. At median follow-ups of 14.5 months and 25.2 months, the incidence of brain metastases in the durvalumab/placebo arms was 5.5%/11.0% and 6.3%/11.8%. In a retrospective review of the prospective, randomized ECOG 1505 trial, adjuvant bevacizumab and chemotherapy did not improve OS compared to those receiving post-operative chemotherapy only, but this agent was associated with a reduction in the incidence of brain metastases [66]. Additionally, in a recent pilot trial, pembrolizumab (10 mg/kg every 2 weeks) resulted in a 29.7% response rate of untreated NSCLC brain mets with a PD-L1 of 1% or greater but no response in five patients with PD-L1 < 1% [67]. Furthermore, in a single-arm, prospective trial of upfront carboplatin/pemetrexed in patients with brain metastases due to lung adenocarcinoma, conventional chemotherapy resulted in a CR/PR of 40% [68].

Other trial results have suggested that IO can cross the blood–brain barrier for patients with untreated, asymptomatic brain metastases from non-small-cell lung cancer. A pilot trial of upfront atezolizumab, carboplatin, and pemetrexed was given to 40 patients accrued from 11 institutions. Patients were noted to have brain metastases from non-squamous, non-small-cell lung cancer without ALK or EGFR genetic alterations. After 4–6 cycles of CT-IO, pemetrexed plus atezolizumab could be given for a maximum of 2 years or until progressive disease. Median intracranial progression-free survival was 6.9 months, with only five patients experiencing intracranial progressive disease. Five patients experienced a complete intracranial response. Objective response rates intracranially and extracranially were 42.7% and 45.0%, respectively. Of note, this trial allowed patients with neurologic symptoms to use up to 4 mg of decadron daily to control their symptoms; 55% (22 patients) used decadron prior and during therapy, with 16 patients receiving the maximum dose of 4 mg of decadron daily. Interestingly, the intracranial/extracranial responses were numerically higher in those receiving decadron (50.0%/52.6%) as compared to those not being treated with corticosteroids (38.9%/44.4%). Likewise, decadron had no effect on OS [69]. Of note, in metastatic disease, it was recognized that a baseline use of >10 mg of prednisone or its equivalence was associated with poorer outcomes in patients with NSCLC who were treated with PD(L)1 inhibition [70]. This trial with brain metastases suggested that the anti-carcinogenic effects of IO may not be as readily inhibited by corticosteroids. Due to its hypothesized and proven ability to overcome immunosuppression in the TME, anti-angiogenic therapy may allow for higher doses of decadron for patients with brain metastases without interfering with the efficacy of IO. Furthermore, by reducing tumor-related edema, anti-angiogenic agents may allow the decadron doses to be tapered more readily. Recently, 171 patients with brain metastases (20 previously treated with radiotherapy) were assessed post hoc from three prospective trials (Keynote 021, -189, and -407) that randomized patients to chemo/pembrolizumab vs. chemotherapy alone. Although prognosis was worse for patients with untreated, stable, or treated brain metastases regardless of treatment group, the benefit of combination therapy was noted for all endpoints (PFS, OS, RR, and duration of response) and all PD-L1 groups (<1%, 1–49%, and ≥50%). The median OS and PFS for the patients with brain metastases (N = 105) treated by chemo/immunotherapy were 18.8 and 6.9 months compared to 7.6 and 4.1 months for those receiving chemotherapy alone (N = 66), respectively [71].

As noted in the introduction, anti-angiogenic agents may also improve the cancer-eliminating effect of IO. Because VEGF has been known to prevent dendritic cell maturation and modulate inhibitor checkpoints on CD8 + T cells in tumors, targeted antiangiogenic agents may also have immunostimulatory effects [30,72]. Additionally, it has been hypothesized and proven that antiangiogenic pharmaceutical agents may act synergistically with IO [27,36].

Therefore, it is felt that a combination of upfront AA, IO, and conventional chemotherapy (AIC) may cause intracranial responses to possibly allow for the conversion of patients needing whole-brain radiation (traditional lesion size ≥3.5 cm or >4 lesions) to radiosurgery treatment alone and/or to decrease the size of lesions amenable for SRS/SRT while providing enhanced local and distal intracranial control. Of interest, it was first noted in 2007 that bevacizumab decreased clinical and radiographic radiation-related necrosis in patients with primary brain tumors while reducing steroid dependency, likely due to preventing capillary leakage and brain edema [73]. The effectiveness of bevacizumab to ameliorate clinical and radiographic necrosis while decreasing steroid dependency has subsequently been confirmed in patients with brain metastases in other investigations [74,75]. Therefore, the administration of anti-angiogenic therapy in patients suffering from brain metastases may not only help overcome the immunosuppressive TME, but it may also help to reduce the steroid dose, which will allow immunotherapy to work better.

Although bevacizumab has been used successfully with CT and IO, its use is limited to only NS-NSCLC. However, Ramucirumab can be combined with chemotherapy or IO and can be used in all NSCLC histologies. As noted above, Ramucirumab and Pembrolizumab have been successfully combined and has shown promising results in patients previously treated with ICI [48]. Additionally, Ramucirumab (10 mg/kg) was combined successfully with carboplatin (AUC = 6) and paclitaxel (200 mg/m^2^) over 21 days in a multi-institutional phase II trial. The efficacy was noted to be similar to the PFS and OS with the addition of bevacizumab to chemotherapy noted in ECOG 4599, and there were no unexpected toxicities and no ≥grade 3 hemoptysis [76]. Moreover, ramucirumab was also combined with Pembrolizumab in 92 patients with gastric/gastroesophageal, NSCLC, and urothelial carcinoma in a phase 1a/1b study. In this study, 27 patients with lung cancer received pembrolizumab 200 mg and ramucirumab 10 mg/kg every three weeks, and there was a 30% response rate [77]. Additionally, ramucirumab was successfully combined with durvalumab (a human immunoglobulin G1 kappa (IgG1κ) monoclonal antibody that blocks the interaction of programmed cell death ligand 1 (PD-L1) with the PD-1) in a phase Ib trial. Forty-three lung cancer patients received ramucirumab 10 mg/kg and durvalumab 1125 mg every three weeks, with no unexpected toxicities [78]. Ivonescimab (bispecific antibody directed against PD-1 and VEGF-A) plus chemotherapy were combined and demonstrated promising efficacy in a single-arm, phase 2 trial in patients with EGFRmut NSCLC whose disease had progressed after previous TKI, and it was associated with a 48.9% response rate and median PFS of 12 months [79]. As noted previously, ivonescimab can be used in all lung cancer histologies [49]. Furthermore, the shorter half-lives of both ramucirumab (13.4 days) [80] and ivonescimab (6–7 days) [81] compared to bevacizumab (20 days) [82] may make them safer agents to use in patients with NSCLC, especially those with brain metastases.

We propose the following strategy for the upfront treatment of NSCLC-related brain metastases for conversion from requiring whole-brain radiation to localized stereotactic treatments for patients with extensive intracranial disease or to decrease the volume of brain metastases requiring stereotactic radiation in order to make the treatment safer and more effective. This investigation would not be for the treatment of those suffering from leptomeningeal disease. The main endpoint of the proposed study is the intracranial benefit (stable disease, complete response (CR), and partial response (PR)) at 6 months by RANO-Brain metastases criteria [83].

The schema for this treatment would be as shown in Figure 3:

Patients in this proposed study would undergo surgical resection of any large or symptomatic brain metastases and be assigned to standard radiation followed by standard concurrent IO and CT. Whole-brain radiation would be given if lesions were greater than 3.5 cm or if there were 10 or more brain metastases. For those requiring whole-brain radiation, hippocampal avoidance (HA-WBRT) would be given to those without hippocampal involvement, and memantine would be used for all to limit neurologic sequelae. CT would be histology-specific, i.e., platinum/pemetrexed for NS-NSCLC or platinum/gemcitabine [84], or a non-histologic-specific regimen such as carboplatin/paclitaxel could be used. In the experimental arm, upfront AA, CT, and IO would be given concomitantly. After the first four cycles of platinum-based CT/IO +/− AA in both arms, patients would then start maintenance IO with or without maintenance pemetrexed (NS-NSCLC only). Because of their shorter half-lives, better safety profiles, and applicability to all NSCLC histologies, ramucirumab and ivonescimab would work better with this strategy. Ivonescimab may be ideal for this situation because of its short half-life and its combined AA and IO properties. After 6 months of therapy on the experimental arm, all lesions 5 mm or less would not be taken into consideration for radiation due to the likelihood that they were controlled, and all other lesions would be taken into account for radiotherapy treatment. In the experimental arm after 6 months of therapy, whole-brain radiation and stereotactic radiosurgery would be given by the criteria above.

Secondary endpoints could include the following:Comparison of systemic and intracranial response as well as maximum responses by RECIST 1.1;Physical functioning—bilateral hand grip strength by hand held dynamometer [85] and timed get up and go (TUG) test to evaluate muscle strength and ability to ambulate [86];Neurologic functioning by Montreal Cognitive Assessment (MoCA) to assess neurocognitive function at baseline and follow-up visits. This tool offers the advantage of being a short but sensitive assessment resistant to biases from the cognitive fatigue anticipated in this patient cohort [87];Intracranial progression-free survival at 6 and 12 months;CTCAEv5.0 for recording neurologic as well as other toxicity;Correlation of intracranial responses with tumor-infiltrating lymphocytes in the stroma > 10%, tumor mutational burden, VEGFR2, VEGFA, and PD-L1 in the baseline tumor samples;The need for radiation and/or neurosurgical intervention during the first 6 months in the different trial arms;Volumetric assessment of the five largest brain metastases per person at the beginning of the trial and at 6 months in both study arms;Assessment of the number of new brain metastases in both arms at 6 months;Determination of whether patients requiring whole-brain radiation can have their tumors successfully reduced in size and number to permit radiosurgery in the experimental AIC arm.

### 2.4. How Can Anti-Angiogenic Therapy and Immunotherapy Be Combined for the Treatment of Locally Advanced, Unresectable NSCLC?

Immediately after ECOG 4599 [32] demonstrated that bevacizumab added to concurrent paclitaxel–carboplatin was demonstrated to have an OS benefit versus the same platinum doublet chemotherapy, many trials were launched to investigate the role of bevacizumab in locally advanced disease (Table 2) [88,89,90,91]. The detrimental techniques that would not be used for a modern trial are delineated in red, bolded print in this table.

As can be seen, all these trials used relatively crude radiation therapy fields that were given with 3-dimensional conformal radiation, and two of the trials treated clinically uninvolved lymph nodes with prophylactic radiation [88,90], which would not be entertained with more modern radiotherapeutic techniques. Additionally, the trial by Socinski et al. used 74 Gy of radiation, which was found to be associated with an OS decrement [92] as compared to chemo/radiotherapy of 60 Gy in a phase 3 trial. Because the local failure rates of lung cancer can are approximately 40%, modern radiation techniques have abandoned treating clinically uninvolved lymph nodes and use IMRT, which has been demonstrated to result in less morbidity by decreasing the dose to cardiac/lung tissue, resulting in lower rates of pneumonitis [93]. Two of these trials used concomitant bevacizumab [88,90] and, along with the poor radiation techniques, resulted in tracheo-esophageal fistulas. Of note, the trial by Patel et al. only had one grade 5 toxicity in 1 of 33 patients, which is an acceptable rate of complications for stage 3 NSCLC, especially with 3D conformal radiation. Importantly, this trial gave bevacizumab only after clinical esophagitis from the concurrent CT-RT had resolved. Strangely, only 39 of 62 patients completed CT-RT followed by two cycles of CT, due largely to progression and/or toxicity. It should be noted that all four of these trials were conducted prior to the Pacific Trial, which noted an unprecedented OS of 47.5 months and PFS of 16.9 for those patients who had not progressed after CT-RT and who received consolidative durvalumab [94]. For both the low-risk NSCLC patients in the trial by Wozniak et al. [89] and the patients treated in the trial conducted by Patel et al. [91], the median PFS/OSs in patients completing the consolidative bevacizumab were equivalent and/or exceeded the PFS/OS of the patients receiving consolidative durvalumab in the Pacific Trial and were noted to be 38/46 and 14.9/42.9, which are much higher than the PFS/OS of conventional chemo/radiotherapy of 9.8–12 mn/25.0–28.7, as demonstrated in past trials [92,95].

Because of the activity of CT, AA, and bevacizumab in patients with liver and brain tumors as well as in the mutations that usually do not respond to immunotherapy, we feel that AIC therapy should be re-explored in patients with locally advanced NSCLC lung cancer by giving this therapy before and after concurrent chemoradiation using IMRT directed to gross disease onlyusing 60 Gy in 30 fractions. A design for this trial can be seen in Figure 4. It is important to remember that this therapy should be given to the appropriate candidates and should only be used for those who are not experiencing hemoptysis or those patients regularly receiving anti-coagulants. Furthermore, all squamous cell carcinomas or centrally located mediastinal masses (<3 cm from the carina) should undergo an MRI or contrast-enhanced CT scan within 21 days of randomization to exclude major blood vessel or airway invasion. Once again, we believe that either ramucirumab and/or ivonescimab would be best for this trial design. Both PFS and OS can be assessed as primary endpoints.

### 2.5. Can Concurrent Immunotherapy, Chemotherapy, and Anti-Angiogenic Therapy Be Beneficial for Patients with EGFR-Mutant Lung Cancer (EGFRmut) Who Progress After Tyrosine Kinase Inhibitors and Amivantamab/Chemotherapy?

Recently, amivantamab and chemotherapy +/− Lazertinib has been proven to both increase PFS and intracranial PFS as compared to chemotherapy alone in patients with EGFRmut lung cancer who have progressed on Osimertinib [96]. However, the median PFS for only amivantamab–chemotherapy and amivantamab–Lazertinib–chemotherapy were only 6.3 and 8.3 months. Therefore, another a third-line strategy is needed.

Both CT-IO and AIC strategies have been used, as noted in patients with EGFRmut NSCLC who have failed previous treatment with tyrosine kinase inhibitors. Table 3 compiles those studies that were published in the peer-reviewed literature. Although Keynote 789 [97] and Checkmate 722 [98] did not show significant differences in their primary endpoints (dual PFS/OS in Keynote 789 and PFS in Checkmate 722) as compared to CT alone, the Orient-31 [99] trial noted a significant PFS benefit when sintillimab was added to CT. Post hoc PFS subgroup analyses of Checkmate 722 demonstrated that there was a possible improved PFS outcome for those who were only treated with one previous line of TKI and/or those with sensitizing mutations (exon 19 deletions; exon 21 L858R mutations). Of note, Checkmate 722 noted that around 84% of patients received only one previous line of TKI, but this information was not available in the other two trials. Although cross-trial comparisons are perhaps not the most valid way of examining the data from these trials, the three trials had similar percentages of patients with exon 19 deletions, exon 21 L858R mutations, other mutations, and never smokers. Although PD-L1 status and T790M mutations were largely unknown in the ORIENT-31 trial, subgroup analysis did not demonstrate any clear benefit of PD-L1% in Checkmate 722 or Keynote 789, and there was only a slight detriment to PFS and OS in patients who had a T90M mutation. What is clear from Table 3 is that PFS in the chemotherapy-alone arm in Orient-31 trial is noticeably less (4.3 months) than that seen in the Keynote 789 (5.5 months) and Checkmate 722 (5.4 months) and brings up the question of whether the PFS benefit of the CT-IO arm of the Orient-31 trial was due to an underperforming control arm. However, all three trials using the AIC strategy had a significantly higher PFS than the CT alone arm, with PFSs of 7.2 vs. 4.3 months with sintillimab/IBI305 (a bevacizumab biosimilar)/CT vs. CT in Orient-31 [99], 8.48 vs. 5.62 months with atezolizumab/bevacizumab/CT vs. CT in the ATTLAS trial [100], and 7.1 vs. 4.8 months with ivonescimab/CT vs. CT in the Harmoni-A trial [81].

Furthermore, in a meta-analysis [101], the AIC regimen was found to yield the best PFS benefit as compared to other therapeutic strategies in patients with metastatic EGFRmut NSCLC who progressed on EGFR TKI. In comparison to CT alone, the HRs for PFS were AIC (HR = 0.54 (0.44–0.67)), IO monotherapy (HR = 1.73, (1.30–2.29)), and IO-CT (HR = 0.77 (0.67–0.88)). The meta-analysis confirmed that AIC yielded a much better PFS than the CT-IO approach (HR = 0.71 (0.59–0.85)) and anti-angiogenic–CT strategy (HR = 0.76 (0.58–1.0)). However, it was noted that AIC yielded higher risks of any-grade and grade 3 or worse adverse events over IO-CT and CT. Therefore, the AIC regimen will likely have its benefits in the properly selected patients. We feel that AIC would be beneficial as a treatment option for those EGFRmut tumors that progress on TKI therapy.

## 3. Summary

Our article reviews the strategy of combining anti-angiogenic therapy with chemotherapy and immunotherapy for NSCLC patients with de novo metastatic disease and those patients with EGFR-mutated NSCLC who have failed previous TKI, and it explores the possibility of using this regimen for patients with stage III unresectable disease and those with brain metastases. Because the previous stage III studies failed due to the administration of bevacizumab concurrently with chemo/radiation and/or using poor radiation techniques (toxically high doses, 3D conformal RT, and treatment of elective nodes), we feel that AIC crucially needs to be explored with modern radiation therapy techniques. We feel that the AIC therapy should be re-explored now that we know how to prevent toxicity with this regimen and that it could be a very efficacious therapy for selected patients with locally advanced, unresectable NSCLC. AIC will likely be beneficial in all disease settings because it is active for mutations recalcitrant to immunotherapy and is active for both prevention and treatment of brain and lung metastases. It should be noted that there are many multi-targeted TKIs that have anti-angiogenic properties, but due to their failure in the first line [102] and in the previously treated populations [102,103,104], we did not think that including anti-angiogenic TKIs would be beneficial in this review.

## 4. Conclusions

This review demonstrates the potential benefits of anti-angiogenic therapy in combination with chemotherapy and immunotherapy (AIC) as a viable strategy for treating EGFR-mutant, TKI-resistant, and de novo metastatic disease in NSCLC patients.

## Figures and Tables

**Figure 1 cancers-16-04207-f001:**
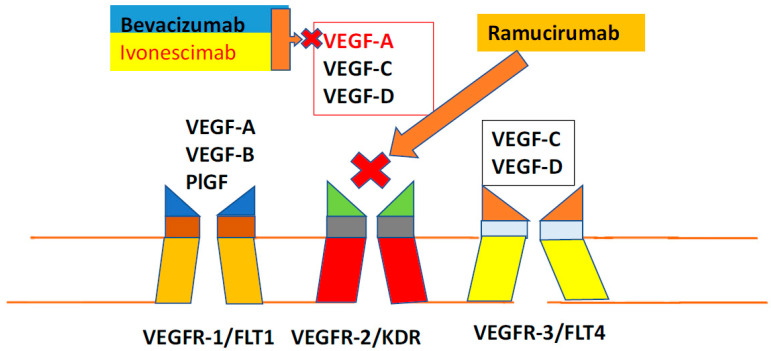
The VEGF family of proteins and their associated receptors as well and the actions of the angiogenic inhibitors.

**Figure 2 cancers-16-04207-f002:**
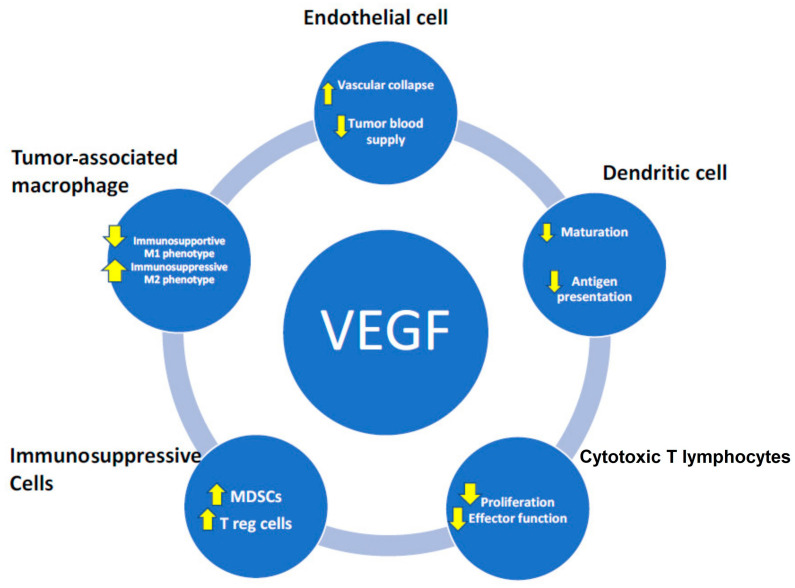
VEGF effect modulation on various cell types in the tumor microenvironment.

**Figure 3 cancers-16-04207-f003:**
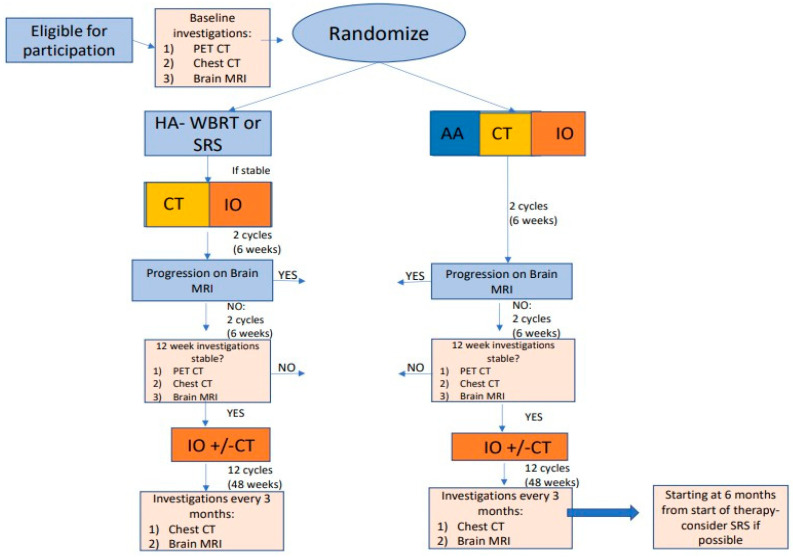
Schema for proposed Brain Metastases trial. AA = anti-angiogenic therapy; CT = chemotherapy; IO = immunotherapy; SRS = stereotactic radiosurgery; HA-WBRT = hippocampal-avoidance whole-brain radiotherapy.

**Figure 4 cancers-16-04207-f004:**
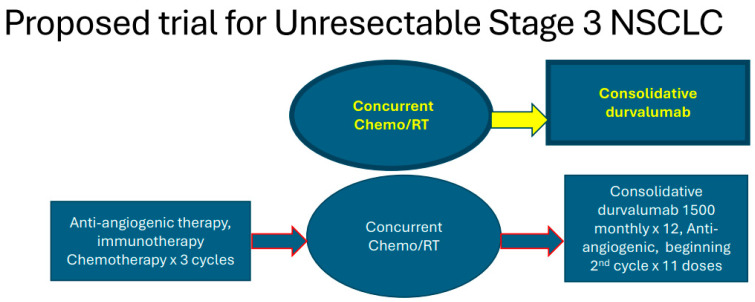
Proposed trial for locally advanced non-small cell lung cancer (RT = radiotherapy).

**Table 1 cancers-16-04207-t001:** Intracranial control and toxicity rates for patients receiving stereotactic radiosurgery (SRS) alone for brain metastases from the four prospective, randomized trials comparing SRS vs. SRS and whole-brain radiation therapy (WBRT).

Institution	Patients	Median Follow-Up (Months)	Patient #	% NSCLC Patients	Local Control	Distal Control	% Receiving Salvage WBRT	Grade 3–4 Toxicity	Symptomatic Necrosis
MD Anderson [53]	Newly diagnosed 1–3 mets	9.5	30	53.0%	67%- 1 year	45%	33%	10% (3/30)	6.67% (2/30)
Multi-institution U.S. [54]	Newly diagnosed 1–3 mets, <3 cm	7.2	111	72.1%	72.8%- 1 year	69.9%	17.7%	10.8% (12/111)	4.5%
EORTC [55]	Newly diagnosed 1–3 brain mets, ECOG PS 0–2	40 months surviving patients	100	52%	69%- 2 year	48%- 2 year	31%	8%	1%
Multi-institution Japan [56]	Newly diagnosed 1–4 brain mets, <3 cm	7.8 months	67	66.0%	72.5%- 1 year	36.3%	16.4%	3.0%	1.5%

**Table 2 cancers-16-04207-t002:** Past clinical investigations using bevacizumab with concurrent chemo/radiation in locally advanced NSCLC.

Study	Follow-Up (Median, mn)/ Patient #	Chemotherapy Regimen	Radiation Details	PFS/OS Median	Toxicity Timing
Spigel 2010 Limited stage Small cell Lung cancer [88]	14 mn/29 patients	Carboplatin/Irinotecan Bev 10 mg/kg every 1 and 14 days for 4 cycles, then maintenance with same schedule for up to 6 months. Concurrent bevacizumab with CHEMO/RT	3D RT-cycle 3—61.2 Gy 2.0–2.5 cm Margins, prophylactic nodal RT Ipsi/Contra Hilum/Sup Med/Paratrach and subcarinal nodes		21 patients could be evaluated 3-TE fistulas—5 min into maintenance
Spigel 2010 JCO LA-NSCLC [88]	5 patients	Carbo/pemetrex/Bev 15 mg/kg q 3 × 2 cycles during, break 7 weeks, and × 3 cycles after radiation, then maintenance 15 mg/kg q3 weeks. Concurrent bevacizumab during RT	RT-cycle 1—61.2 Gy Anterior/Posterior ports		2 TE fistulas 34–40 weeks into therapy
Wozniak Clin Lung Cancer 2015 [89] LA-NSCLC High- Risk (Squamous, Hemoptysis, cavitation or adjacent to major blood vessel) Low-risk: All others	26 patients	1. Cisplatin/etoposide with RT, docetaxel, Bev 15 mg/kg × 3 cycles—start 4–7 weeks after RT sequential bevacizumab	3D RT to primary lesion and nodes 64.8 Gy	PFS/OS- 38/46 months low risk; 15/17 months high risk	2 of 11 high-risk—fatal Hemoptysis-squamous tumor with cavitation; centrally located adenocarcinoma 15 low-risk stratum
Socinski JC0 2012 [90]	53 months /45 patients 27% squamous	2 cycles of induction carbo/paclitaxel and bev 15 mg/kg q 3 weeks, Concurrent carbo/paclitaxel/RT/bev 10/kg q 2 weeks × 4, Cohorts 2 and 3 erlotinib—4 days/week, 100 mg and 150 mg during CT/RT/Bev Consolidation erlotinib 150 mg daily and bev 15 mg/kg q 3 weeks—6 cycles	RT, after 2 cycles of induction, gross disease + prophylactic node RT—2 cm beneath clavicular heads, 2 cm beneath the carina, initial fields AP/PA, boost to gross disease post chemotherapy to 74 Gy, 3D RT 75.5% completed RT—74 Gy	10.2/18.1 months	1 TE fistula—3.5 months after combined therapy, 29% grade 3–4 esophagitis
Patel Clin Lung Cancer 2020 [91]	68 patients, but only 39 patients completed chemoRT, 33 consolidation	Carboplatin/paclitaxel CRT, no progression, then 2 cycles of full-dose CT Step 2 Cyclophosphamide ×1, followed by Bev 15 mg/kg q 3 weeks, tecemotide—up to 34 doses of BEV permitted Sequential bevacizumab	RT—66 Gy to all grossly involved tumor and nodes, 3D RT only	14.9/42.7	39 completed step 1 33 pts started step 2—median # cycles 11-1 grade 5 esophageal fistula

RT = radiotherapy; TE = trachea-esophageal; Bev = bevacizumab, 3D = three dimensional, conformal radiation therapy.

**Table 3 cancers-16-04207-t003:** Table of trials comparing chemotherapy–IO and AIC therapy to chemotherapy alone in patients with EGFRmut who failed previous TKI therapy.

Trial Arms	Pt #	Follow-Up Times	% with Exon 19d; Exon 21m; Other	Stratification Factors	% Treated with Osimertinib or Other 3rd-Generation Drug	% T790M Mutations	% Brain Mets	Smoking History	PD-L1 Status <1% 1–49% ≥50%	Trial Endpoints
**Keynote-789 [97]** Platinum/pemetrexed	247	42.0 mn	56.7% 42.0% 0.4%	TPS ≥ 50% Prev Osimertinib East Asia vs. other	47.3%	38.8%	46.9%	34.3% current/former; 65.7% never	51.8%; 22.0% 21.2%	PFS 5.6 mn (Exp) vs. 5.5 mn—NS
Platinum/Pemetrexed/pembrolizumab	245	42.0 mn	57.5% 41.3% 0.4%		39.0%	35.2%	49.0%	33.6% current/former; 66.4% never	45.7% 29.1% 20.6%	OS 15.9 mn (Exp) vs. 14.7%— NS
**Checkmate 722 [98]** Platinum/pemetrexed	150	38.1 mn	54.7% 38.7% 14%	PdL-1- < 1 Brain mets Smoking Previous Osimertinib	24.0%	6.0%	34.0%	37.3% current/former; 62.87% never	40.0% 24.7% 26.0%	PFS 5.6 mn (Exp) vs. 5.4 mn—NS
Platinum/pemetrexed/ nivolumab	144	38.1 mn	55.6% 37.5% 15.4%		22.2%	6.3%	35.4%	38.9% current/former; 61.1% never	37.5% 29.2% 20.8%	
**ORIENT-31 [99]** Cisplatin/pemetrexed	160	14.4 mn	55.5% 38% 7%	Sex Brain mets	37%	Unknown	37%	29% current/former; 71% never	4% 2% 3% 92% (unknown)	
Cisplatin/pemetrexed/sintillimab	158	15.1 mn	54% 39% 7%		34%	Unknown	37%	31% current/former; 69% never	7% 0% 3% 90% (unknown)	PFS 5.5 mn (Exp) vs. 4.3 mn *
Cisplatin/pemetrexed/sintillimab/IBI305	158	12.9 mn	51% 44% 5%		39%	Unknown	37%I BIS	30% current/former; 70% never	3% 2% 2% 94% (unknown)	PFS 7.2 mn vs. 4.3 mn *
**ATTLAS [100]** Platinum/pemetrexed (8.1% ALK)	154	26.1 mn	61.8% 36.8% 1.5%	Mutation type Brain mets	42.6%	29.4%	41.9%	62.2% never; 37.8% current/former	49.0%; Unknown: 16.3%	PFS 8.48 mn (Exp) vs. 5.62 mn *
Atezolizumab/bevacizumab/ carboplatin/paclitaxel (4.6% ALK)	74	26.1 mn	47.6% 51.0% 1.4%		42.9%	34.7%	43.5%	63.0% never; 37.0% current/former	59.8%; Unknown: 16.3%	
**HARMONI-A [81]** Carboplatin/pemetrexed	161	7.89 mn	48.4% 48.4% 15.5%	Prev 3rd gen TKI Brain mets	85.1%	11.2%	23.0%			PFS— 7.1 mn (Exp) vs. 4.8 mn *
Carboplatin/pemetrexed/ivonescimab	161	7.89 mn	57.1% 37.3% 21.7%		86.3%	16.1%	21.7%			

Exon 19d = exon 19 deletion mutations; Exon 21m = exon 21 L858R mutations; Exp = experimental arm of the trial; * = significance difference between trial arms; NS = non-significance between trial arms.

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
