# Peer review of "Evaluation of Anti-Angiogenic Therapy Combined with Immunotherapy and Chemotherapy as a Strategy to Treat Locally Advanced and Metastatic Non-Small-Cell Lung Cancer"

_cancers, 2024, doi:10.3390/cancers16244207_

Round 1
Reviewer 1 Report
Comments and Suggestions for Authors
Referee Report
This manuscript presents a review comparing combinations of anti-angiogenic therapy and immunotherapy, with and without chemotherapy, in the treatment of non-small cell lung cancer. The topic is relevant and timely. However, I have some minor suggestions for improvement:
- Clarification of Article Type: The authors should confirm whether this is a review or a research article. Although it is marked as a research article, it reads more like a topical review.
- Title: The title could be rephrased to better convey the study’s purpose, as the current wording does not clearly reflect the study's objectives.
- Keywords: No keywords are provided in the manuscript. Including relevant keywords will improve the discoverability of the paper.
- Introduction: The Introduction section is too brief. The authors should expand on the background of the study, define medical terminology, and discuss recent advancements. Additionally, it would be helpful to state the aim of this review and explain its significance.
- Exclusion of Surgery and Radiotherapy: Surgery and radiotherapy are not addressed in this study, despite being common treatments for lung cancer. The authors should clarify the rationale for excluding these modalities.
- Figure Captions: Figure 2 lacks a caption, which should be added for clarity.
- Formatting Issues in Tables:
- In Table 1, Row 3, Column 1, “US55” appears with an inconsistent font size.
- Table 2 has a different formatting style from Table 1, which detracts from consistency. Additionally, the red font requires clarification, as does the inconsistent reference style across tables. This issue is also present in Table 3.
- Conclusion: A Conclusion section is missing. Including one would help summarize the findings and emphasize key takeaways.
Author Response
We are sorry for the inconvenience, bellow are the point-by point reply:
- It's a topical review not a research article
- Evaluation of anti-angiogenic therapy combined with immunotherapy and chemotherapy as a strategy to treat advanced non-small cell lung cancer.
- Keywords:
Advanced, non-small cell lung cancer (NSCLC), chemoimmunotherapy, anti-angiogenic, combination
- The following paragraph already been added and highlighted in our response:
The tumor micro-environment (TME) consists of blood and lymphatic vessels, various stromal cells, and resident and infiltrating immune cells, all of which are ensconced in an extracellular matrix. In comparison with nonmalignant tissues, each component of the TME is abnormal and heterogeneous, and this abnormality fuels tumor progression and treatment resistance [1]. Normalizing components of the TME could improve the outcomes of patients treated with standard and emerging anticancer therapies, including CT, RT, molecularly targeted agents, and IO [2].
We feel that the combination of AA therapy, IOs and cytotoxic therapy may be beneficial for non-small cell lung cancer (NSCLC) in the setting metastatic disease without targetable mutations, brain metastases, locally advanced disease, and for recurrent EGFR mutated (EGFRmut) disease. We will review the clinical research for locally advanced and metastatic NSCLC and discuss how we can improve our therapeutic armamentarium in these 4 areas.
- There is a limited role for surgery or radiation (except for palliation) in treating advanced non-small cell lung cancer. The standard of care for advanced non-small cell lung cancer is a systemic treatment.
- VEGF effect modulation on various cell types in the tumor microenvironment.
- Corrected and highlighted in the response.
8.Already been added and highlighted in the response, we added the following:
This review demonstrates the potential benefits of anti-angiogenic therapy added in combination with chemotherapy and immunotherapy (AIC) as a viable strategy for treating EGFR mutant, TKI resistant and de novo metastatic disease in NSCLC patients.
------------------------------------------------------------------
The manuscript already been updated and uploaded

Reviewer 2 Report
Comments and Suggestions for Authors
The manuscript “Combined Anti-Angiogenic Therapy, Immunotherapy With or Without Chemotherapy and Its Incorporation into Difference Strategies for the Treatment of Non-Small Cell Lung Cancer” submitted to the journal Cancers is deal with the discussion how anti-angiogenic therapies may potentiate the anti-carcinogenic effects of immunotherapy of Non-Small Cell Lung Cancer. The topic discussed is very important for increasing the long-term survivals and improving treatment of Non-Small Cell Lung Cancer.
I would like to make some comments:
1. It is advisable to shorten the manuscript title.
2. Add the title of the figure to Figure 2, and also sign the cell type near the fragment with the designations “Proliferation Effector function”.
3. In the note to Table 1, indicate the decoding of the abbreviations.
4. In Table 2, unify the designations of patients.
5. Please, design all tables in the same color scheme.
6. Bring the conclusions in the Annotation into line with the conclusions set out in the Summary section.
Author Response
Reviewer 2
- We did rephrase the title; the new title is:
Evaluation of anti-angiogenic therapy combined with immunotherapy and chemotherapy as a strategy to treat advanced non-small cell lung cancer.
- It's been added and highlighted in the response.
- It's been added and highlighted, we added the following to decode the abbreviations:
Stereotactic Radiosurgery (SRS), whole brain radiation therapy (WBRT).
- We unified the designations of the patients to table 2 (unified and highlighted).
5.Corrected and highlighted.
- The conclusion been added to the summary .
-----------------------------------------------------------------
I uploaded the revised version of the manuscript
Round 2
Reviewer 1 Report
Comments and Suggestions for Authors
The authors' responses file is the manuscript file so the reviewer cannot work on this revision.
Round 3
Reviewer 1 Report
Comments and Suggestions for Authors After reviewing the authors’ responses and the revised manuscript attached to this email, I am satisfied with the modifications and corrections made by the authors in response to my comments. They have addressed all my concerns, and I have no further questions regarding this manuscript for publication. Comments on the Quality of English LanguageNo comment.